# Integrating CT Radiomics and Clinical Features to Optimize TACE Technique Decision-Making in Hepatocellular Carcinoma

**DOI:** 10.3390/cancers17050893

**Published:** 2025-03-05

**Authors:** Max Masthoff, Maximilian Irle, Daniel Kaldewey, Florian Rennebaum, Haluk Morgül, Gesa Helen Pöhler, Jonel Trebicka, Moritz Wildgruber, Michael Köhler, Philipp Schindler

**Affiliations:** 1Clinic for Radiology, University of Münster, 48149 Münster, Germany; 2Department of Internal Medicine B, University of Münster, 48149 Münster, Germany; 3Department of General, Visceral and Transplant Surgery, University of Münster, 48149 Münster, Germany; 4Department of Radiology, LMU Munich, 80336 Munich, Germany

**Keywords:** HCC, TACE, radiomics, integrated diagnostics

## Abstract

Many people with advanced liver cancer cannot have surgery, so they often undergo a procedure called transarterial chemoembolization. However, it can be difficult to choose the best treatment technique for each patient. In this research, the authors proposed a way to combine information from computed tomography scans with a patient’s medical details to predict how well each type of chemoembolization method might work. Using special statistics, this study aimed to guide doctors toward the most appropriate treatment for each person in a model-based approach. This could improve how patients are selected for therapy and inspire further studies towards more personalized approaches to liver cancer treatment.

## 1. Introduction

Hepatocellular carcinoma (HCC) is the most common primary liver malignancy and represents a major global health burden, with increasing incidence and mortality rates worldwide [1]. While early-stage HCC may be amenable to curative interventions such as surgical resection, transplantation, or ablation, many patients presented with intermediate or advanced disease, for which locoregional therapies, particularly transarterial chemoembolization (TACE), play a pivotal role in treatment [2,3,4]. TACE involves the selective catheter-based delivery of chemotherapeutic and embolic agents to induce tumor ischemia and necrosis, and has become a standard treatment option for patients with preserved liver function and unresectable multifocal disease [2,4]. In addition, TACE is also used as a bridging therapy prior to liver transplantation (LT) to control local tumor growth and/or down-stage tumor burden, allowing patients to meet the eligibility criteria for LT [5].

Despite its widespread use, outcomes following TACE are heterogeneous, reflecting the complexity of patient- and tumor-specific factors as well as the variability inherent in procedural techniques and lack of standardization [6]. Several TACE techniques have emerged combining various embolics as well as chemotherapeutic agents, including conventional TACE (cTACE), drug-eluting bead TACE (DEB-TACE), and degradable starch microsphere TACE (DSM-TACE), each with different pharmacokinetic profiles, safety considerations, and therapeutic effectiveness [6,7,8,9]. Choosing the best treatment for an individual patient remains a critical yet unresolved challenge, as current clinical practice guidelines lack consensus on the optimal TACE technique [2,8,10]. Selection of a certain TACE technique is often driven by institutional preferences and the experience of the interventional radiologist, rather than necessarily reflecting patient-specific factors.

In recent years, radiomics has gained momentum as a promising tool to improve personalized treatment selection in oncology [11]. Radiomics involves the high-throughput extraction of quantitative imaging features from standard scans, such as computed tomography (CT), to characterize intra-tumoral heterogeneity and underlying tumor biology [12]. When combined with clinical and laboratory data, radiomic signatures may help predict therapeutic responses, stratify patients by risk, and guide selection of optimal interventions. In the context of HCC, emerging evidence suggests that radiomics-based models can improve prognostication, assess tumor vascularity and biology, and potentially predict response to TACE [13,14,15,16].

This study aimed to integrate CT radiomics with key clinical variables to develop a decision framework for selecting the most appropriate TACE technique for HCC. By building a predictive model that incorporates both imaging-based features and patient-specific factors, we aimed to improve the precision and personalization of TACE interventions for HCC.

## 2. Materials and Methods

### 2.1. Study Design and Patient Selection

This study was conducted as a retrospective, single-center, observational study at a tertiary care academic liver center in accordance with the tenets of the Declaration of Helsinki. The study was approved by the Ethics Committee of the University of Münster and Westphalia-Lippe (ID: 2020-495-f-S). Informed consent was not obtained from patients due to the retrospective nature of this study.

All patients diagnosed with HCC who underwent at least one TACE procedure at our center between 2008 and 2021 (*n* = 277) were reviewed. All TACE procedures were approved by an interdisciplinary gastrointestinal tumor board. Inclusion criteria were: (1) age ≥ 18 years; (2) diagnosis of HCC confirmed by imaging criteria according to the European Association for the Study of the Liver (EASL) guidelines [2]; (3) eligibility for TACE due to BCLC intermediate stage B, bridging or down-staging to liver transplantation (LT), or other stages when patients were unsuitable for resection, LT, local ablation, or systemic therapy; (4) availability of pre-TACE contrast-enhanced CT images and follow-up imaging at 4–6 weeks post-TACE. Exclusion criteria included TACE in treatment history, additional treatment with second TACE or other treatments within 4–6 weeks of first TACE, and inadequate image quality.

Based on these criteria, a total of 151 patients were finally included in the study. Patients were divided into three cohorts according to the performed TACE procedure: 33 received cTACE, 69 received DEB-TACE, and 49 received DSM-TACE. A flowchart of patient selection is shown in Figure 1.

### 2.2. TACE Procedures

In this study, all TACE procedures were performed by board-certified interventional radiologists with over four years of experience and training in all three TACE techniques. The choice of TACE approach was determined by the interventional radiologist performing the procedure. cTACE, DEB-TACE, and DSM-TACE were conducted in accordance with CIRSE standards of practice and as described previously [8,17].

Briefly, the procedure started with retrograde 5 F trans-femoral access, followed by pump-based angiography of the mesenteric artery and celiac trunk, and hepatic artery. Angiograms were performed at a flow rate of 4 mL/s with a 20 mL volume of contrast agent, to delineate the tumor’s arterial supply. A coaxial microcatheter system (2.0–2.4 F) was then advanced as selectively as possible to the tumor. Selectivity was categorized as follows: (1) superselective, where only tumor vessels were visualized in the control angiogram; (2) selective, where tumor vessels and one non-tumor sub-segmental branch were contrasted; (3) unselective, where tumor vessels and multiple non-tumor branches (segmental or lobar) were contrasted. Cone-beam CT (CBCT) was utilized as needed to verify microcatheter positioning or to identify additional or variant tumor feeders.

Drug delivery was carried out using a 1–3 mL syringe in a fluoroscopic-controlled fashion. For cTACE, a water-in-oil emulsion comprising 50 mg of doxorubicin and 10 mL of lipiodol (1:2 ratio) was used. DEB-TACE employed 100 µm Embozene microspheres (Varian Medical Systems, Palo Alto, CA, USA) loaded with 50 mg of doxorubicin per the manufacturer’s instructions. DSM-TACE utilized 50 µm degradable starch microspheres (Embocept, Sirtex Medical Europe, Bonn, Germany) mixed with 50 mg of doxorubicin. The endpoint of each procedure was the achievement of blood flow stasis in the tumor’s feeding vessels. Final confirmation of embolization material accumulation within treated HCC lesions was performed using CBCT. Peri-procedural medications included intravenous administration of 7.5 mg piritramide for analgesia and 4 mg ondansetron for anti-emetic purposes.

### 2.3. Data Collection and Follow-Up

All patient and procedural data, including laboratory values and follow-up/survival information, were retrospectively retrieved from electronic medical records and the picture archiving and communication system (PACS). Patients were evaluated at baseline and 4 to 6 weeks post-TACE according to the modified Response Evaluation Criteria in Solid Tumors (mRECIST) using diagnostic CT or MRI if CT was contraindicated [18]. Only HCC lesions that were subsequently treated with TACE were analyzed according to mRECIST. Lesions treated with TACE may differ from “target lesions” defined by the standards of mRECIST, exemplarily if only one (out of two definable mRECIST “target lesions”) was addressed by the first TACE procedure due to dose limits. Treatment response was categorized as response (complete or partial) and non-response (stable disease, progressive disease). This approach involves measuring the greatest diameter of the remaining arterial-enhancing (viable) portion of each TACE-treated lesion. Complete response is defined as the disappearance of arterial enhancement in all target lesions, while partial response requires at least a 30% reduction of the sum of viable diameters relative to the baseline measurement. Progressive disease occurs if there is at least a 20% increase of the sum of these diameters, compared to the smallest sum recorded. Stable disease encompasses changes that do not meet the thresholds for partial response or progressive disease. Figure 2 illustrates the response assessment schematically.

### 2.4. Imaging and Radiomics Analysis

All patients underwent contrast-enhanced multiphasic CT scans within four weeks prior to TACE using standardized protocols. The imaging protocol encompassed the acquisition of late arterial phase images between 20 and 35 s following contrast injection or a well-timed arterial phase (based on bolus tracking), as well as portal venous phase images between 60 and 70 s after injection. CT images underwent quality control and independent evaluation by two abdominal radiologists with eight years of experience in liver imaging. These radiologists were blinded to the patients’ clinical and laboratory data. The analysis focused on the identification of typical HCC lesions by their arterial hypervascularity and subsequent washout on delayed images according to the LI-RADS criteria [20]. For descriptive statistics, all lesions that met the radiologic criteria for HCC (equivalent to LI-RADS 4 and 5) were analyzed, and the size and number of all HCC lesions were recorded to calculate the hepatic tumor burden and define the Milan criteria. Portal vein invasion was evaluated.

The arterial phase images were selected for radiomic feature (RF) extraction due to their superior contrast between the hypervascular tumor and the liver parenchyma, as well as their enhanced robustness in semi-automatic segmentation. Semi-automated segmentation by volumetric regions of interest (VOIs) of HCC lesions was performed using the open-source 3D Slicer software package version 5.6.2 [21]. HCC VOIs were defined in the arterial phase according to RECIST 1.1 criteria on a per-patient basis, encompassing the entire tumor volume subsequently treated with TACE and including any intra- and inter-tumoral heterogeneity [22]. Areas of high-density embolic material or artifacts were excluded from the segmented region of interest to avoid RF bias, when necessary, in pre-treated patients. In accordance with the guidelines established by the Image Biomarker Standardization Initiative (IBSI), a total of 91 RFs for HCC lesions corresponding to LI-RADS 4 and 5, with a diameter of ≥1 cm, and subsequently treated with TACE were extracted [23,24]. The features encompassed morphology, intensity-based features, histogram-based features (first-order texture), and gray-level co-occurrence matrix elements (GLCM, second-order texture). To ensure the comparability of the extracted RFs, they were standardized using z-score normalization. The reliability of the selected RFs was assessed by calculating the concordance correlation coefficient (CCC) between two readers. Features with a coefficient between 0.8 and 1 were considered “excellent” and included for further analysis. Figure 3 illustrates the imaging and radiomics analysis workflow.

### 2.5. Development of Best TACE Technique Decision Framework

Data preparation: All analyses were conducted using R (version 4.4.2, R Foundation for Statistical Computing, Vienna, Austria) with the tidyverse, caret, randomForest, purrr, broom, ggplot2, dotwhisker, interactions, dplyr, glmnet, reshape2, and ggalluvial packages. The dataset included a binary outcome variable “Response”, categorized as yes (complete or partial response) or no (stable or progressive disease), and a categorical variable “TACE technique” with three treatment categories: cTACE, DEB-TACE, and DSM-TACE. In addition, multiple baseline clinical, laboratory, and radiomic predictor variables were provided, including demographic factors, liver disease etiology, performance status classifications (ECOG), tumor staging (BCLC), and radiomic features (Appendix A).

Feature selection and model specification: To identify predictors associated with treatment response, we employed a penalized logistic regression approach using elastic net regularization. Prior to modeling, predictors were separated from the response variable, and the TACE technique category was retained as an independent variable. The outcome response was used as the dependent variable in all subsequent modeling steps. The elastic net model was implemented via the glmnet function, which applies both LASSO (L1) and Ridge (L2) penalties in a manner controlled by the tuning parameter alpha. Here, a cross-validated elastic net model was fitted to binomial logistic regression, using cv.glmnet to identify the optimal penalty parameter lambda. The final chosen model leveraged the lambda.min value identified via 10-fold cross-validation. A combination of L1 and L2 regularization, controlled by alpha, was used to simultaneously achieve feature selection (LASSO’s strength) and coefficient shrinkage (Ridge’s strength), thereby improving model interpretability and reducing the risk of overfitting.

Model fitting and coefficient interpretation: Following model fitting, the regression coefficients associated with each predictor were extracted. Coefficients shrunk to zero by the elastic net penalty were considered non-informative and thus excluded from subsequent interpretation. The remaining nonzero coefficients were exponentiated to produce odds ratios, facilitating intuitive interpretation of the direction and magnitude of each predictor’s association with the probability of a response. Predictors with odds ratios deviating substantially from 1 (chosen here as ±5%) were considered significant. These significant predictors were retained for subsequent visualization and modeling steps.

Visualization and interaction effects: Forest plots and bar charts were generated to visualize the distribution of key predictors across TACE techniques and response groups. To investigate the interplay between treatment strategies and selected predictors, a generalized linear model including interaction terms between TACE technique and each significant predictor was fitted. This allowed for the estimation of predicted response probabilities under different treatment modalities and patient profiles. The interactions and ggplot2 packages were employed to generate bar plots illustrating predicted probabilities across TACE techniques for hypothetical or observed patient profiles.

Comparison between recommended and actual treatments: A procedure was implemented to generate individualized treatment recommendations based on predicted probabilities. For each patient, the probability of response was estimated under each TACE technique, and the technique with the highest predicted probability was considered the recommended option. These recommendations were then compared to the actual techniques received by the patients to quantify concordance and to explore patterns of disagreement.

Performance and concordance analysis: To assess the alignment between recommended and actual treatments, a confusion matrix was created, comparing the advised TACE technique versus the one administered. Alluvial (Sankey) diagrams were used to visualize treatment switches and agreements. This enabled a quantitative evaluation of how often the model-driven recommendations would have differed from clinical choices, potentially providing insights into areas for clinical improvement or model refinement.

### 2.6. Statistical Analysis

Descriptive statistical analysis was performed using SPSS Statistics version 29 (SPSS Inc., Chicago, IL, USA). Data are presented as total number and percentage, mean and standard deviation (SD), median and interquartile range (IQR), or 95% confidence interval (CI), as appropriate. Statistical analysis was performed using Chi-Square test with Bonferroni post hoc test for categorical and ANOVA with Sidak post hoc test for continuous variables. *p* values < 0.05 were considered statistically significant.

## 3. Results

### 3.1. Patient Characteristics and Response Assessment

The patient characteristics of the study population are shown in Table 1. There were no significant differences in age, gender distribution, BCLC stage, underlying liver disease etiology, or catheter position selectivity between the three TACE cohorts. A total of 87 patients (57.6%) received TACE for bridging/down-staging, while 49 patients (32.5%) initially met the Milan criteria with no differences in the subgroups. Thus, data shows homogeneity and comparability between groups. The vast majority of HCCs were diagnosed radiologically and only a minority received an additional histological diagnosis, either by biopsy (9.9%) or resection prior to TACE (5.3%), again with no subgroup differences.

At 4–6 weeks follow-up, DEB-TACE was associated with significantly higher response rates (complete or partial response) than cTACE [DEB, *n* = 39 (56.5%) vs. cTACE, *n* = 10 (30.3%); *p* = 0.046]. In addition, neither DEB-TACE was statistically superior to DSM-TACE [response rate, *n* = 23 (46.9%)], nor was DSM-TACE statistically superior to cTACE.

### 3.2. Multivariable Regression Analysis of Predictors of TACE Response

In the multivariable analysis using an elastic net regularized logistic regression, three variables were identified as independent predictive factors for post-TACE response (Figure 4).

The corresponding coefficient weights are presented in Table 2. In descending odds ratio order, the factor associated with increased response probability was the radiomic feature “Contrast” (odds ratio: 5.80), while BCLC stage B (odds ratio: 0.92) and viral hepatitis (odds ratio: 0.74) as underlying liver disease were associated with decreased response probability. The “Contrast” feature is part of the gray level co-occurrence matrix category. It quantifies the intensity contrast between a pixel and its neighbor across the entire image, reflecting the amount of local variation present. A higher contrast value indicates greater intensity differences between neighboring pixels, suggesting a more heterogeneous texture. Conversely, a lower contrast value indicates a more homogeneous texture with minimal intensity variation.

### 3.3. Development and Evaluation of a Prediction Model Based on Independent Predictors Identified by Regression Analysis

Further analyses were performed to explore variations in significant predictors across TACE techniques and response categories. Interaction models were utilized to assess how changes in predictor values modified predicted response probabilities under each TACE technique. In particular, a logistic regression model that included interaction terms between significant predictors (“Contrast”, BCLC stage, etiology of liver disease) and TACE technique demonstrated that the relative benefit of certain TACE techniques depended on specific patient and imaging characteristics (Figure 5).

To evaluate the model’s recommendations against real-world treatment decisions, predicted probabilities of response for each TACE technique were generated for individual patients. The modality with the highest predicted response probability was identified for each case, and these recommendations were compared to the actual treatments received. Figure 6 provides an overview of the frequency and types of discrepancies between recommended TACE techniques and those actually performed. In only 51 patients (33.8%) did the model-based recommendation match the actual TACE technique used. In 100 patients (66.2%), a different TACE technique was associated with a higher probability of response according to our model. The highest mismatch rate was in the combination DSM-TACE performed/DEB-TACE recommended (*n* = 24, 24.0%); the lowest mismatch rate was in the combination cTACE performed/DSM-TACE recommended (*n* = 7, 7.0%). The highest concordance rate was observed in patients with DEB-TACE (*n* = 36, 52.2%).

## 4. Discussion

In this study, we integrated radiomic features extracted from pre-TACE CT scans with clinical variables to develop a decision framework for selecting the most effective TACE technique in HCC. Our results confirmed the heterogeneity of patient outcomes after TACE and underscore the complexity of guiding therapy selection based solely on traditional clinical parameters [25]. Current guidelines include TACE in intermediate-stage HCC and bridging/down-staging scenarios [2,4,8,10]. However, there is no consensus regarding the optimal TACE technique, as institution- and interventionalist-related factors have historically guided technique selection [2,8,10].

First, our study highlights that DEB-TACE is associated with better response rates compared to cTACE, as evidenced by higher rates of complete and partial responses according to mRECIST criteria at 4–6 weeks after TACE. This finding is consistent with several previous studies suggesting improved pharmacokinetic profiles, better local drug delivery, and more controlled embolization with drug-eluting beads [26,27,28]. Although DSM-TACE did not show statistically significant superiority over cTACE or DEB-TACE, it is noteworthy that degradable starch microspheres have been proposed to achieve transient but effective embolization while allowing for subsequent retreatment strategies and potentially improved tolerability [29,30]. Moreover, the lack of significant difference between DEB-TACE and DSM-TACE challenges the notion of a single “best” TACE technique, with recent (multicenter) studies showing lower objective response rates (ORR), similar disease control rates (DCR) and survival rates, and lower liver toxicity of DSM-TACE compared to other techniques [17,29,31]. This shows that additional patient-specific factors and tumor biology should be considered when selecting the optimal TACE technique.

Second, a multivariable model identified three independent predictors of TACE response, including BCLC stage B, underlying viral hepatitis etiology, and a radiomic feature (“Contrast”). As expected, BCLC stage B is associated with a lower predictive probability of response compared to stage A, while the numbers of cases in stages C and D were too small to be statistically significant. Patients with BCLC A were intentionally included in the study model, with TACE used as bridging therapy for patients awaiting liver transplantation to help control tumor growth and maintain eligibility within transplant criteria [5], or occasionally for individuals who have significant comorbidities—such as severe cardiac or pulmonary disease—or who have declined surgery [32]. Finally, certain tumor characteristics, including difficult-to-access locations near major vessels, may make resection or ablation less safe or effective, making TACE a viable therapeutic alternative.

An interesting finding is the negative association of viral hepatitis etiology with treatment response. Although the design of this study does not allow us to identify the exact pathophysiological pathway, several mechanisms associated with chronic viral hepatitis, particularly hepatitis B (HBV) and hepatitis C (HCV), have been linked with reduced efficacy of TACE [33,34,35]. Persistent viral infection drives chronic inflammation in the liver, fostering an immunosuppressive microenvironment that can undermine the cytotoxic effects of TACE [36]. Chronic hepatitis often leads to progressive fibrosis and cirrhosis, which subsequently impairs liver function and limits the degree of embolization and chemotherapy doses that can be safely administered [37]. Ongoing viral replication may exacerbate tumor angiogenesis, altering intra-tumoral vascular architecture and potentially reducing TACE-induced ischemia [38,39]. Finally, emerging evidence suggests that virally mediated molecular pathways may support more aggressive tumor phenotypes, further reducing the ability of TACE to induce adequate necrosis [35]. Such findings support the notion that not only tumor stage and radiographic phenotype, but also the patient’s underlying liver disease profile, should be considered [40]. Elucidating the molecular and microenvironmental underpinnings of poor response in viral hepatitis-associated HCC may lead to better therapeutic stratification, possibly incorporating combination therapies with targeted agents or immunotherapies that have shown promise in certain HCC subsets [38,39]. By integrating radiomics, we have moved beyond simple morphological criteria to include quantitative imaging biomarkers that may reflect underlying tumor biology and vascularity [41,42]. Among the various features, “Contrast” emerged as an independent predictor. This feature likely captures the heterogeneity in tumor vascularity and texture patterns, which have been previously linked to TACE responsiveness [25]. Importantly, a radiomic feature such as “Contrast” is an objective, quantifiable parameter supporting the otherwise very subjective and rater-dependent assessment of heterogeneity in vascularity.

Third, our analysis revealed a substantial discrepancy between the TACE technique predicted to yield the highest probability of response and the TACE procedure actually performed. Although the model suggested that a different TACE technique could improve the probability of response in the majority of patients, real-world decisions were consistent with the model’s recommendation in only about one-third of cases. Institutional preferences and expertise play an important role, as different centers and individual interventional radiologists often develop familiarity with particular embolic materials and procedural techniques. Patient comorbidities and performance status also play a role, with liver function, cardiovascular status, and the toxicity profiles of various agents guiding clinical decision-making. The reason for the high proportion of recommended switches from DSM-TACE to DEB-TACE seems to be that this study focused on response but not on liver toxicity, liver damage, or altered hemodynamics (e.g., transjugular intrahepatic portosystemic shunt, TIPS). The interventional radiologist may have taken these factors into account and therefore selected DSM-TACE, which is thought to have less toxicity [17,29,31]. For this study, however, it was not useful to consider side effects or liver toxicity, as these are overall rare with TACE. Ultimately, our results suggest that objective, data-driven models could complement and potentially refine clinical judgment, helping to standardize practice and guide individualized therapy.

The integration of a radiomics-based predictive model into clinical workflows has the potential to shift the paradigm of TACE technique selection from empirically driven “one-size-fits-all” approaches to personalized interventions [40,43]. By stratifying patients according to their predicted likelihood of response to different TACE techniques, clinicians can better identify those who would benefit most from cTACE, DEB-TACE, or DSM-TACE. Such a strategy could improve outcomes and reduce unnecessary procedures.

However, several limitations should be acknowledged. The retrospective design and single-center nature may limit generalizability. In addition, the study focused primarily on short-term response rather than long-term survival or LT success rate. Further TACE cycles (common practice) or non-TACE intermediate therapies were not analyzed or predicted. A prediction based on the premise that a certain TACE technique or other intermediate therapy was used shortly before the procedure could be completely different, underscoring the importance of prospectively collecting survival and recurrence data in future studies to refine and validate the predictive value of our model over longer follow-up periods. In addition, although our model identified factors correlated with response, external validation in a prospective, multicenter cohort is needed to ensure the generalizability, reproducibility, and clinical utility of these findings. We do not include the histopathologic differentiation of the HCC in our analysis, as these were not well-processed in the retrospective setting. More importantly, the goal of this study was to develop a predictive imaging biomarker-based model pre-TACE that can be used to predict response in a real-world scenario before TACE is actually performed, with the ultimate goal being the improvement of both patient and treatment selection and matching patients with the right TACE technique. In the future, the inclusion of a broader panel of image-derived features (including advanced imaging features like peritumoral enhancement, MRI-based radiomics, and emerging molecular imaging techniques) and the integration of standardized tissue sampling and molecular tumor characteristics (e.g., gene expression profiles) may improve the robustness and predictive accuracy of these models [44,45,46,47]. Methodologically, the use of elastic net regression facilitated both variable selection and model regularization. Elastic net penalization blends L1 (LASSO) and L2 (Ridge) penalties, offering advantages over simple LASSO or Ridge methods alone [48]. It efficiently handles high-dimensional predictor spaces, especially from radiomics, and reduces the risk of overfitting by shrinking coefficient estimates [49,50]. However, weaknesses remain: the choice of tuning parameters can influence which predictors are ultimately selected, and certain complex nonlinear relationships may not be fully captured by a standard logistic model. By capturing complex and nonlinear image patterns that may be overlooked by conventional radiomics approaches, neural networks and convolutional architectures have the potential to provide more accurate disease characterization [51,52]. In addition, these models have the capacity to integrate data from multiple sources, including imaging (e.g., CT/MRI), clinical parameters, and molecular markers, thereby offering a more holistic approach to prediction [51,52]. Furthermore, deep learning methods have the capability to automate feature extraction and selection, which has the potential to reduce operator dependence and increase reproducibility in radiomics-based studies, but at the cost of reduced interpretability [53].

## 5. Conclusions

This study illustrated the predictive value of both imaging-derived radiomic features and key clinical factors for TACE response in patients with HCC, and highlights a significant gap between model-based recommendations and real-world TACE technique selection. These findings underscore the need for data-driven decision support tools to optimize TACE technique selection, ultimately advancing the field toward more personalized and effective locoregional therapies for HCC.

## Figures and Tables

**Figure 1 cancers-17-00893-f001:**
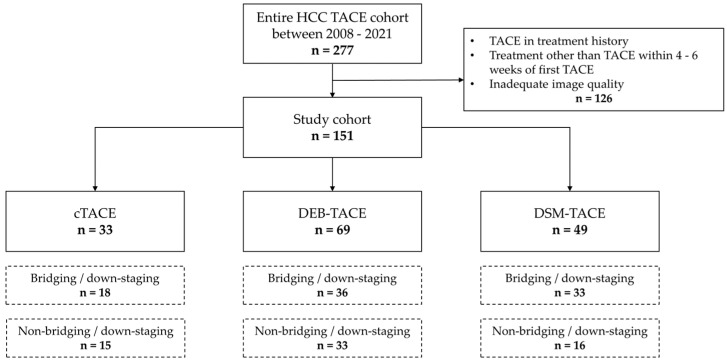
Flowchart of patient selection. Abbreviations: cTACE, conventional transarterial chemoembolization; DEB-TACE, drug-eluting bead transarterial chemoembolization; DSM-TACE, degradable starch microsphere transarterial chemoembolization; HCC, hepatocellular carcinoma; TACE, transarterial chemoembolization.

**Figure 2 cancers-17-00893-f002:**
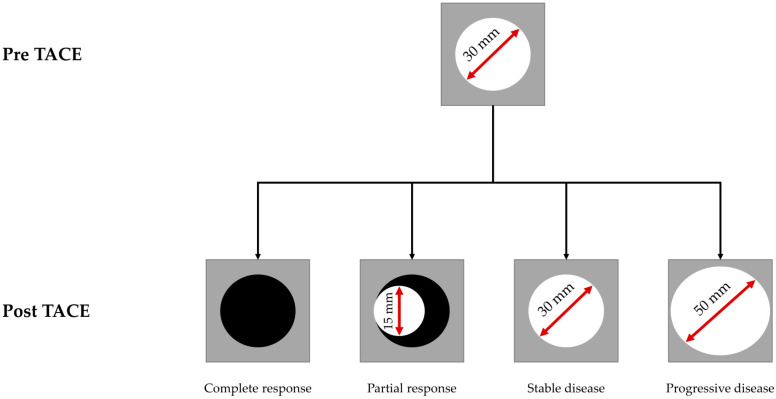
Schematic illustration of the mRECIST criteria for response assessment after TACE in HCC. The arterial phase hyper-enhancing part of the tumor is shown in white. Adapted to Dioguardi Burgio et al. [19]. Abbreviations: HCC, hepatocellular carcinoma; mRECIST, Modified Response Evaluation Criteria in Solid Tumors; TACE, transarterial chemoembolization.

**Figure 3 cancers-17-00893-f003:**
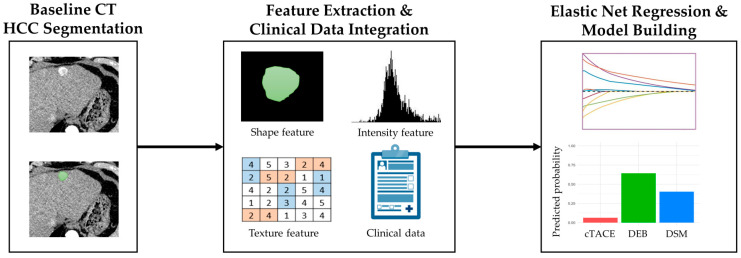
Scheme of imaging and radiomics analysis workflow. After baseline segmentation of HCC was subsequently treated with TACE (green segmentation), radiomic features including shape, intensity, and texture were extracted and integrated with clinical data. Feature selection and model building were performed using elastic net regression. Bar charts were then generated to illustrate predicted probabilities across TACE techniques for hypothetical or observed patient profiles. Abbreviations: HCC, hepatocellular carcinoma; TACE, transarterial chemoembolization.

**Figure 4 cancers-17-00893-f004:**
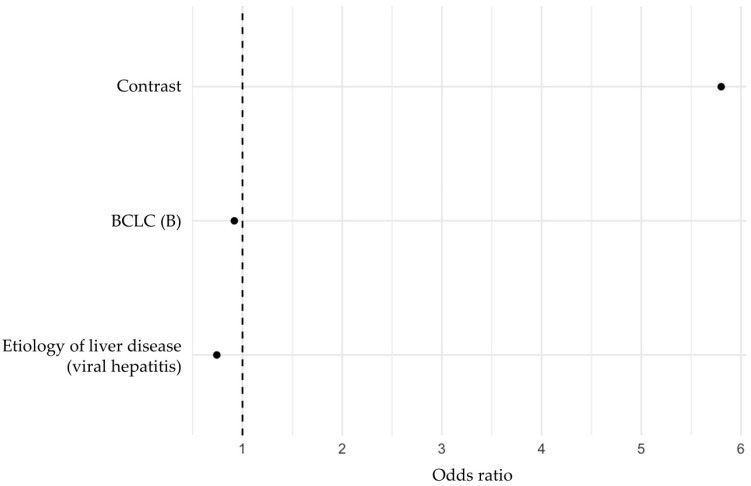
Multivariable analysis of variables selected by elastic net regularized logistic regression for predicting TACE response. Abbreviations: BCLC, Barcelona Clinic Liver Cancer staging.

**Figure 5 cancers-17-00893-f005:**
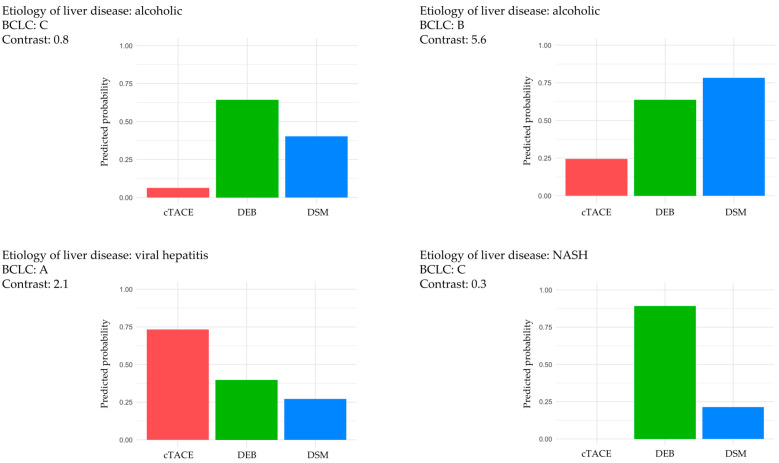
For demonstration purposes, four patients with different values of the identified predictors were selected. The different values of the predictors etiology of liver disease, BCLC stage, and the radiomic feature “Contrast” result in different prediction probabilities of treatment response depending on the TACE technique to be selected. Abbreviations: BCLC, Barcelona Clinic Liver Cancer staging; cTACE, conventional transarterial chemoembolization; DEB, drug-eluting bead; DSM, degradable starch microsphere; NASH, non-alcoholic steatohepatitis; TACE, transarterial chemoembolization.

**Figure 6 cancers-17-00893-f006:**
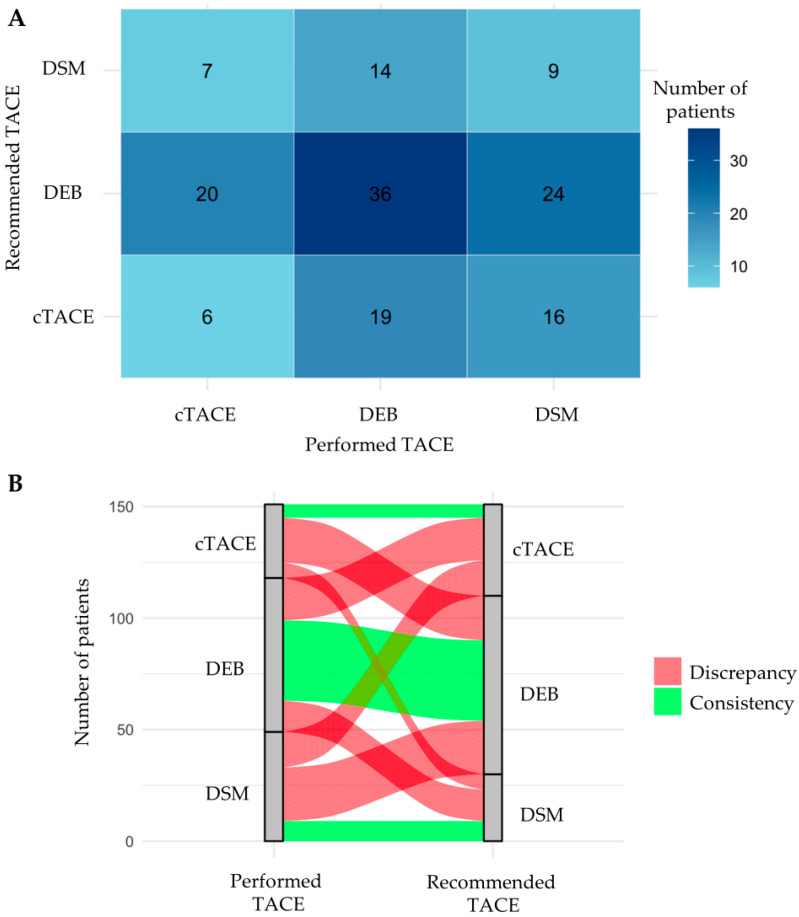
A confusion matrix (**A**) and a Sankey diagram (**B**) summarize the frequency and nature of discrepancies between model-based recommended and performed TACE techniques. Abbreviations: cTACE, conventional transarterial chemoembolization; DEB, drug-eluting bead; DSM, degradable starch microsphere; TACE, transarterial chemoembolization.

**Table 1 cancers-17-00893-t001:** Patient characteristics and response assessment.

	Total	cTACE	DEB	DSM	** *p* **
**Clinical and laboratory data**					
N	151 (100)	33 (21.9)	69 (45.7)	49 (32.5)	
Sex					
Male	120 (79.5)	27 (81.8)	52 (75.4)	41 (83.7)	0.508
Female	31 (20.5)	6 (18.2)	17 (24.6)	8 (16.3)	
Age (years), median (IQR)	64.0 (12.0)	60.0 (10.0)	63.0 (17.0)	65.0 (10.0)	0.325
Etiology of liver disease					
Alcoholic	63 (41.7)	17 (51.5)	26 (37.7)	20 (40.8)	0.882
Viral hepatitis	40 (26.5)	9 (27.3)	17 (24.6)	14 (28.6)	
Biliary disease	1 (0.7)	0 (0)	1 (1.4)	0 (0)	
NASH	14 (9.3)	2 (6.1)	7 (10.1)	5 (10.2)	
Toxic	1 (0.7)	0 (0)	0 (0)	1 (2.0)	
Autoimmune hepatitis	1 (0.7)	0 (0)	1 (1.4)	0 (0)	
Hemochromatosis	1 (0.7)	0 (0)	1 (1.4)	0 (0)	
Cryptogenic	30 (19.9)	5 (15.2)	16 (23.2)	9 (18.4)	
ECOG					
0	76 (50.3)	14 (42.4)	34 (49.3)	28 (57.1)	0.406
1	61 (40.4)	13 (39.4)	30 (43.5)	18 (36.7)	
2	12 (7.9)	5 (15.2)	5 (7.2)	2 (4.1)	
3	2 (1.3)	1 (3.0)	0 (0)	1 (2.0)	
BCLC					
A	79 (52.3)	14 (42.4)	33 (47.8)	32 (65.3)	0.322
B	51 (33.8)	13 (39.4)	24 (34.8)	14 (28.6)	
C	13 (8.6)	3 (9.1)	8 (11.6)	2 (4.1)	
D	8 (5.3)	3 (9.1)	4 (5.8)	1 (2.0)	
Diagnosis of HCC by liver biopsy Previous therapy	15 (9.9)	4 (12.1)	7 (10.1)	4 (8.2)	0.634
Resection	8 (5.3)	2 (6.1)	4 (5.8)	2 (4.1)	0.111
Systemic	7 (4.6)	2 (6.1)	4 (5.8)	1 (2.0)	0.575
Local					
RFA	2 (1.3)	1 (3.0)	1 (1.4)	0 (0)	0.350
MWA	1 (0.7)	0 (0)	0 (0)	1 (2.0)	
TARE	11 7.3)	2 (6.1)	6 (8.7)	3 (6.1)	
PEI	4 (2.6)	0 (0)	1 (1.4)	3 (6.1)	
INR, median (IQR)	1.2 (0.2)	1.2 (0.3)	1.2 (0.3)	1.1 (0.2)	0.168
Bilirubin (mg/dL), median (IQR)	1.0 (0.9)	1.2 (1.5)	0.9 (0.9)	1.0 (0.8)	0.449
ALT (U/L), median (IQR)	41.5 (38.3)	39.0 (42.5)	39.0 (33.0)	38.0 (33.0)	0.274
AST (U/L), median (IQR)	57.5 (42.5)	51.0 (41.0)	58.0 (48.0)	50.0 (42.0)	0.329
ALP (U/L), median (IQR)	128.5 (97.3)	130.0 (101.0)	122.0 (95.0)	117.0 (85.0)	0.226
GGT (U/L), median (IQR)	140.0 (193.5)	84.0 (168.5)	173.0 (151.0)	117.0 (291.0)	0.490
AFP (ng/mL), median (IQR)	16.4 (167.6)	8.4 (133.3)	17.6 (277.6)	17.8 (67.5)	0.486
**Imaging features and procedural data**					
Indication for TACE					
Bridging/down-staging	87 (57.6)	18 (54.4)	36 (52.2)	33 (67.3)	0.239
Non-bridging/down-staging	64 (42.4)	15 (45.5)	33 (47.8)	16 (32.7)	
Hepatic tumor burden					
0–25%	133 (88.1)	27 (81.8)	60 (87.0)	46 (93.9)	0.309
26–50%	16 (10.6)	6 (18.2)	8 (11.6)	2 (4.1)	
>50%	2 (1.3)	0 (0)	1 (1.4)	1 (2.0)	
Portal vein invasion	9 (6.0)	4 (12.1)	3 (4.3)	2 (4.1)	0.239
Within Milan criteria	49 (32.5)	6 (18.2)	22 (31.9)	21 (42.9)	0.105
Sum of target lesion diameter (mm), median (IQR)	39.1 (31.5)	34.4 (19.9)	44.5 (36.5)	37.0 (30.3)	0.137
Catheter application position					
Unselective	73 (48.3)	14 (42.2)	31 (44.9)	28 (57.1)	0.102
Selective	51 (33.7)	14 (42.4)	23 (33.3)	14 (28.6)	
Superselective	27 (17.9)	5 (15.1)	15 (21.7)	7 (14.3)	
**Follow-up**					
Response at 4–6-week follow-up					
CR/PR	72 (47.7)	**10 (30.3) ^a^**	**39 (56.5) ^b^**	23 (46.9) ^a,b^	**0.046**
SD/PD	79 (52.3)	**23 (69.7) ^a^**	**30 (43.5) ^b^**	26 (53.1) ^a,b^	

Values denote n (%) or median (IQR) ANOVA for continuous variables, Chi-Square test for categorial variables (Bonferroni post hoc test: Values with the same letter superscripted do not vary significantly). Abbreviations: AFP, alpha-fetoprotein; ALP, alkaline phosphatase; ALT, alanine transaminase; AST, aspartate transaminase; BCLC, Barcelona Clinic Liver Cancer (stage); CR, complete response; cTACE, conventional transarterial chemoembolization; DEB, drug-eluting bead; DSM, degradable starch microsphere; ECOG, Eastern Cooperative Oncology Group (performance status); GGT, gamma-glutamyl transferase; HCC, hepatocellular carcinoma; INR, international normalized ratio; MWA, microwave ablation; NASH, non-alcoholic steatohepatitis; PD, progressive disease; PEI, percutaneous ethanol injection; PR, partial response; RFA, radiofrequency ablation; SD, stable disease; TACE, transarterial chemoembolization; TARE, transarterial radioembolization.

**Table 2 cancers-17-00893-t002:** Coefficient weights of variables for response analysis.

Parameter	Regression Coefficient	Odds Ratio
Contrast	1.75808882	5.8013394
BCLC (B)	−0.08530392	0.9182332
Etiology of liver disease (viral hepatitis)	−0.29781417	0.7424393

## Data Availability

The data that support the findings of this study are available from the corresponding author, P.S., upon reasonable request.

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
