# Peer review of "Integrating CT Radiomics and Clinical Features to Optimize TACE Technique Decision-Making in Hepatocellular Carcinoma"

_cancers, 2025, doi:10.3390/cancers17050893_

Round 1
Reviewer 1 Report
Comments and Suggestions for Authors
Table 1: 79 patients (52.3%) are BCLC A, 49 patients (32.5%) initially met the Milan criteria, and 15 patients had a history of local regional therapy. Please explain (although this is retrospective) why the BCLC A patients received non-curative embolization therapy rather than resection or ablation.
Author Response
Reviewer 1
Comment: Table 1: 79 patients (52.3%) are BCLC A, 49 patients (32.5%) initially met the Milan criteria, and 15 patients had a history of local regional therapy. Please explain (although this is retrospective) why the BCLC A patients received non-curative embolization therapy rather than resection or ablation.
Response: We appreciate your observation regarding the inclusion of BCLC A patients in a study of transarterial chemoembolization (TACE). Although surgical resection and/or local ablation are typically recommended for early-stage hepatocellular carcinoma (HCC) and are also the first-line approach in our center, in real-world clinical settings, there are several reasons why some BCLC A patients undergo TACE instead:
- Bridging to liver transplantation (LT): Some patients were listed for LT but had to wait for a suitable donor organ. During this waiting period, TACE was employed to prevent tumor progression and keep the tumor within transplant criteria as recommended by national and international guidelines.
- Patient comorbidities or refusal of surgery: Certain patients had significant comorbidities (e.g., severe cardiac or pulmonary disease) or declined surgery. In such cases, TACE was considered a feasible alternative.
- Tumor location or morphology not amenable to resection/ablation: Some BCLC A patients had tumors in difficult-to-reach locations (e.g., near major vessels) that limited the efficacy or safety of ablation or resection.
In addition to the Methods section, where we have already described the inclusion criteria in detail, we have addressed this point again in the revised Discussion.
Reviewer 2 Report
Comments and Suggestions for Authors
This study explores the integration of CT radiomics and clinical features to optimize the selection of transarterial chemoembolization techniques for hepatocellular carcinoma. Using a retrospective analysis of 151 patients who underwent CTACE, DEV-TACE, or DSM-TACE, the researchers applied elastic net-regularized logistic regression to identify key predictors of treatment response. They found that the radiomics feature "contrast", BCLC stage B and viral hepatitis etiology were independent predictors of response. The model-generated recommendations for TACE technique selection differed from actual clinical decision in 66.2% of cases, suggesting a potential for improved patient-technique matching. The study highlights the promise of personalized TACE selection based on imaging and clinical data but emphasizes the need for external validation and consideration of log-term outcomes to refine decision-making.
Your study provides an innovative approach to optimizing TACE treatment selection in HCC by integrating radiomics and clinical variables. Below are a few suggestions for further improvement:
1. The study notes that the model-based recommendation differed from the actual TACE technique used in 66.2% of cases. It would be useful to discuss potential reasons why clinicians may have opted for different treatments such as institutional preferences, experience levels, and patient comorbidities.
2. The study focuses primarily on short-term response (4-6 weeks post TACE). If feasible, incorporating survival or recurrence data could strengthen the clinical applicability of the model.
3. Given the study's single-center design, external validation with a multicenter dataset could further establish the generalizability of the findings.
4. The study identifies viral hepatitis as a factor associated with decreased response probability. Expanding on potential biological mechanisms such as inflammatory micoenvironment and fibrosis could enhance the discussion.
5. Given the limitations acknowledged in the manuscript, discussing how AI or deep learning could further refine predictive modeling in TACE selection would be a valuable addition.
Author Response
Reviewer 2
Comment 1: The study notes that the model-based recommendation differed from the actual TACE technique used in 66.2% of cases. It would be useful to discuss potential reasons why clinicians may have opted for different treatments such as institutional preferences, experience levels, and patient comorbidities.
Response: We agree that multiple real-world factors influence TACE technique selection. In the revised Discussion section, we have expanded our explanation to include:
- Institutional preferences and expertise: Different centers and individual interventional radiologists may have familiarity with, or preference for, specific TACE materials and methods.
- Patient comorbidities and performance status: Clinicians often tailor TACE based on liver function, cardiovascular comorbidities, and the potential toxicity profiles of various embolic agents.
These factors can result in treatment decisions that deviate from model-based recommendations that are solely treatment response, highlighting the need for integrative decision support tools.
Comment 2: The study focuses primarily on short-term response (4-6 weeks post TACE). If feasible, incorporating survival or recurrence data could strengthen the clinical applicability of the model.
Response: We acknowledge that long-term outcomes such as overall survival, recurrence-free survival, and post-transplant outcomes could strengthen the clinical applicability of the model. Due to the retrospective nature of this study and variations in patient follow-up intervals, especially with numerous and varied non-TACE intermediate therapies, data on long-term survival or recurrence were not standardized and comparable across patients. Our study rather focusses on the prediction of the treatment outcome of the first performed TACE procedure, a measure that is an important milestone for further therapeutic decision in clinical scenarios.
We have emphasized in the Discussion the importance of prospective collection of survival and recurrence data in future studies to refine and validate the predictive value of our model over extended follow-up periods.
Comment 3: Given the study's single-center design, external validation with a multicenter dataset could further establish the generalizability of the findings.
Response: We fully agree that external validation in multiple centers and diverse patient populations would strengthen the reliability and generalizability of our model. In the revised Discussion, we have underscored the need for prospective, multicenter collaborations to replicate our findings and refine the predictive algorithms for broader clinical use.
Comment 4: The study identifies viral hepatitis as a factor associated with decreased response probability. Expanding on potential biological mechanisms such as inflammatory microenvironment and fibrosis could enhance the discussion.
Response: Thank you for pointing this out. We have expanded this paragraph in the Discussion elaborating on potential mechanisms by which chronic viral hepatitis could reduce TACE efficacy, including:
- Inflammatory microenvironment and fibrosis: Persistent inflammation and fibrosis might alter tumor perfusion and the local immune response.
- Molecular pathways: Viral hepatitis can induce oncogenic pathways that may lead to more aggressive or treatment-resistant tumor phenotypes.
These factors likely influence the heterogeneity in TACE outcomes for patients with viral hepatitis–related HCC.
Comment 5: Given the limitations acknowledged in the manuscript, discussing how AI or deep learning could further refine predictive modeling in TACE selection would be a valuable addition.
Response: We appreciate this suggestion. In the revised Discussion, we have expanded the subsection on the potential of advanced machine learning techniques, including deep learning, to:
- Capture nonlinear relationships: Neural networks and convolutional models may discern complex image patterns not easily captured by conventional radiomics approaches.
- Integrate multi-modal data: Future models could combine imaging data (CT/MRI), clinical parameters, and molecular markers to improve prediction of treatment response, complications and survival.
- Automate feature extraction: Deep learning methods can automate the entire feature extraction and selection pipeline, potentially reducing operator dependence and improving reproducibility.
Reviewer 3 Report
Comments and Suggestions for Authors
There are some comments.
1. It would be better to expand pathologic data (e.g., tumor size, gross type, histology) and correlate with treatment response to evaluate whether specific histologic characteristics influence the efficacy of different TACE techniques.
2. Providing a more detailed explanation of CT radiomics methodology would be better.
3. It would be better to clarify treatment response assessment criteria and imaging metrics and add imaging examples to improve clarity.
Comments on the Quality of English LanguagePlease check English grammar and spelling
For example, Kryptogenic -> cryptogenic
Author Response
Reviewer 3
Comment 1: It would be better to expand pathologic data (e.g., tumor size, gross type, histology) and correlate with treatment response to evaluate whether specific histologic characteristics influence the efficacy of different TACE techniques.
Response: We acknowledge the value of detailed pathologic and histologic information in refining predictive models. However, due to the retrospective design and the fact that not all patients underwent resection or biopsy (e.g., those who received TACE as bridging or palliative therapy or when the diagnosis of HCC was confirmed by imaging criteria according to the EASL guidelines), comprehensive histologic data were limited.
More importantly, the goal of this study was to develop a predictive imaging biomarker-based framework pre-TACE that can be used to predict response probabilities in a real-world scenario before TACE is actually performed, with the ultimate goal of improving both patient and treatment selection and matching patients with the right TACE technique. We agree that including histopathology may further improve those models, yet with aim mentioned beforehand we focused on sets of biomarkers that can be obtained before the time of TACE or transplantation. We now point this out clearly in the Discussion section and note in the Limitations section that a prospective study with standardized tissue sampling would allow a deeper histologic correlation with TACE outcomes.
Comment 2: Providing a more detailed explanation of CT radiomics methodology would be better.
Response: We have expanded the imaging and radiomics analysis subsection to provide readers with a clearer view of our radiomics workflow. We additionally now provide a workflow scheme of the radiomics methodology for better illustration (Figure 3).
Comment 3: It would be better to clarify treatment response assessment criteria and imaging metrics and add imaging examples to improve clarity.
Response: We have included more details in the Methods section about how we defined treatment response for the HCC lesions treated with TACE using the modified Response Evaluation Criteria in Solid Tumors (mRECIST), which specifically accounts for the viable (enhancing) portion of the tumor post-locoregional therapy.
We have also updated the text to clarify our imaging metrics, specifying the arterial-phase measurements used for segmentation and subsequent radiomics.
Regarding imaging examples, we have added representative figures to illustrate how response assessment, tumor segmentation and radiomic feature extraction were performed (Figure 2 and Figure 3).
Additional Editorial and Linguistic Revisions
We have performed a thorough proofreading of the manuscript to ensure that grammar and spelling adhere to academic US English standards.
The term “kryptogenic” has been corrected to “cryptogenic” where applicable.
Minor typographical errors have been corrected throughout the text.
Round 2
Reviewer 3 Report
Comments and Suggestions for Authors
The manuscript was well-revised.
There are some comments.
It would be better to describe the number of patients who underwent resection and biopsy.
Please edit the references to fit the journal's format.
Please check English grammar and spelling.
Fore xample, labaratory -> laboratory
Author Response
Reviewer 3
Comment: It would be better to describe the number of patients who underwent resection and biopsy.
Response: We have expanded the first Results subsection and Table 1 to include data on the number of patients who underwent resection and those whose HCC was histologically confirmed after biopsy.
Comment: Please edit the references to fit the journal's format.
Response: We have edited the references to fit the journal's format.
Comment: Please check English grammar and spelling. For example, labaratory -> laboratory.
Response: Again, we have performed a thorough proofreading of the manuscript to ensure that grammar and spelling adhere to academic US English standards. The term “labaratory” has been corrected to “laboratory” where applicable.